# A First-in-Human Trial to Evaluate the Safety and Immunogenicity of a G Protein-Based Recombinant Respiratory Syncytial Virus Vaccine in Healthy Adults 18–45 Years of Age

**DOI:** 10.3390/vaccines11050999

**Published:** 2023-05-18

**Authors:** Xin Cheng, Gan Zhao, Aihua Dong, Zhonghuai He, Jiarong Wang, Brian Jiang, Bo Wang, Miaomiao Wang, Xuefen Huai, Shijie Zhang, Shuangshuang Feng, Hong Qin, Bin Wang

**Affiliations:** 1Advaccine Biopharmaceuticals Suzhou Co., Ltd., Suzhou 215000, China; 2Key Laboratory of Medical Molecular Virology (MOE/NHC/CAMS), School of Basic Medical Science, Fudan University, Shanghai 200000, China

**Keywords:** respiratory syncytial virus, vaccine, safety, immunogenicity

## Abstract

Background: With the enormous morbidity and mortality caused by respiratory syncytial virus (RSV) infections among infants and the elderly, vaccines against RSV infections are in large market demand. Methods: We conducted a first-in-human (FIH), randomized, double-blind, placebo-controlled dose escalation study to evaluate the safety and immunogenicity response of the rRSV vaccine (BARS13) in healthy adults aged 18–45. A total of 60 eligible participants were randomly assigned to receive one of four dose levels or vaccination regimens of BARS13 or placebo at a 4:1 ratio. Results: The mean age was 27.40, and 23.3% (14/60) were men. No treatment-emergent adverse events (TEAEs) led to study withdrawal within 30 days after each vaccination. No serious adverse event (SAE) was reported. Most of the treatment-emergent adverse events (TEAEs) recorded were classified as mild. The high-dose repeat group had a serum-specific antibody GMC of 885.74 IU/mL (95% CI: 406.25–1931.17) 30 days after the first dose and 1482.12 IU/mL (706.56–3108.99) 30 days after the second dose, both higher than the GMC in the low-dose repeat group (885.74 IU/mL [406.25–1931.17] and 1187.10 IU/ mL [610.01–2310.13]). Conclusions: BARS13 had a generally good safety and tolerability profile, and no significant difference in terms of adverse reaction severity or frequency was observed between different dose groups. The immune response in repeat-dose recipients shows more potential in further study and has guiding significance for the dose selection of subsequent studies.

## 1. Introduction

Respiratory syncytial virus (RSV) infection is a major cause of respiratory tract disease in children under 5 years old. It leads to 64 million cases of bronchiolitis and viral pneumonia [1,2,3,4,5] and causes about 200,000 deaths annually [4,6]. A prophylactic treatment using palivizumab, and more recently, nirsevimab [7], can be used to prevent RSV in premature newborns and infants with certain congenital heart defects or bronchopulmonary dysplasia and infants with congenital malformations of the airway. However, antibodies’ economic cost limits their use in infants with identified risk factors residing in the developed world [8]. Besides the huge threat among the pediatric population, RSV infection is now recognized as a significant problem in elderly adults. Attack rates in nursing homes are approximately 5–10% per year, with substantial rates of pneumonia (10–20%) and death (2–5%). Estimates using US health care databases and viral surveillance results over a 9-year period indicate that RSV infection causes approximately 10,000 all-cause deaths annually among persons >64 years of age [9]. Although a vaccine is considered a more economical and effective strategy for preventing RSV-infected disease, no vaccine is currently available. This problem arises from the severity of the pathologic responses induced by vaccination with formalin-inactivated RSV (FI-RSV) to a large extent.

In the 1960s, the FI-RSV vaccine caused severe lung injuries in some pediatric recipients, and two infants died, resulting in a phenomenon that is now called vaccine-enhanced disease (VED) or enhanced respiratory disease (ERD) [10]. Pathological analysis showed that the dead infants had extended peribronchiolitis and alveolitis [11,12,13]. Subsequent studies have associated the formalin-inactivated RSV vaccine (FI-RSV) with a low level of antibody response and CD4+ T priming in the absence of cytotoxic T lymphocytes resulting in a pathogenic Th2 memory response with eosinophils and neutrophils [14], and such exacerbated T cell responses have been associated with immunopathogenesis of RSV in experimental models [15,16]. Further understanding of the virus and VED mechanisms entails a new RSV vaccine design being required [17,18].

As of August 2022, global RSV vaccine development progress shows that only one Astra Zeneca RSV monoclonal antibody for pediatrics has been approved for marketing, 13 trials are at stages Phase II or Phase III, 11 trials are at Phase I, and more than a dozen candidates are still in preclinical phase [19]. More recently, one more AstraZeneca RSV antibody, nirsevimab, has been approved to prevent RSV lower respiratory tract disease in newborns and infants on 15 September 2022 [7]. The majority of phase II/III clinical trials are focused on pediatric and elderly populations. In contrast, the adult population accounts for a minority, and only three trials are being carried out. The RSV vaccine based on F protein as the main target has been considered for RSV vaccine developments, including Pfizer’s and GlaxoSmithKline’s F-protein-based RSV vaccine (for adults and the elderly) in phase III, while Janssen’s adenovirus vaccine and Merck’s RSV anti-F mAbs are also in the race.

Comparatively, the candidate vaccine (BARS13) is based on a recombinant RSV viral G protein (RSV-G), containing two active components in an optimized ratio which are a purified RSV-G (expressed in *E. coli* system), which functions as the antigenic component, and cyclosporine A (CsA), which functions as an immunomodulator and the diluent to reconstitute the RSV-G. The G protein has been selected as the RSV immunogenic candidate as it has more stable neutralizing epitopes that are comparatively independent of its protein structure [20]. G protein functions as an attachment protein during an RSV infection by interacting with the receptor of target cells. A monoclonal Ab against G protein has demonstrated activity in inhibiting an RSV infection in animal models [21]. CsA is a widely used immunosuppressant in organ transplantations and autoimmune diseases. It can induce antigen-specific T regulatory cells (Tregs) to ultimately achieve tolerogenic responses when combined with a protein antigen at a certain ratio and under a certain dose level [22]. In the development of BARS13, CsA was successfully used to generate tolerogenic responses with human PBMCs in vitro [23]. As Treg plays an essential role in the suppression of VED [24], BARS13 was developed using a combination of RSV-G with CsA to induce functional Tregs and a high level of neutralizing anti-RSV antibodies.

Preclinical studies have been performed in mice, rabbits, and rhesus macaque monkeys to investigate the immunological response to BARS13 and protective efficacy from the RSV challenge following immunization. It has been demonstrated that BARS13 not only induces a high level of neutralizing Abs against RSV but also suppresses the exacerbated lung inflammation that occurs in animals vaccinated with either FI-RSV- or G protein-based vaccines after an RSV challenge [25]. Based on these preclinical studies, we sought to test the safety, reactogenicity, and immunogenicity of the BARS13 investigational vaccine when administered intramuscularly (IM) to healthy adult participants aged 18 to 45 years.

## 2. Materials and Methods

### 2.1. Participants and Study Design

We conducted a phase I, first-in-human (FIH), randomized, double-blind, placebo-controlled dose-escalation study in healthy adults at a single center in Australia (Nucleus Network, Melbourne, Australia) from 16 October 2018 to 5 July 2019. The participants received either a single or repeat vaccination schedule and a different RSV-G protein plus CsA dose mixture. Participants, investigators, and laboratory staff were masked to treatment allocation. The primary objective was to evaluate the safety and reactogenicity of BARS13, and the secondary objective was to evaluate the humoral response in terms of immunoglobulin g (IgG) antibody levels to BARS13. The neutralization antibody response and T cell response were evaluated as exploratory objectives. The trial information can be obtained from Clinical Trial Registration (https://clinicaltrials.gov/ct2/show/NCT04851977 (accessed on 19 April 2021)).

Healthy males or females aged 18–45 years with no history of severe allergy or immunosuppressive therapy were screened for eligibility. All participants provided written informed consent before participation. The participants were enrolled and randomized in a 4:1 ratio sequentially using a dose-escalation protocol to receive low-dose BARS13 (one injection of 9.2 µg rRSV-G protein/10 µg CsA to one arm, and one injection of saline/mannitol to the other arm), high-dose BARS13 (one injection of 9.2 µg rRSV-G protein/10 µg CsA per arm) or placebo (one injection of saline/mannitol per arm). Both of the investigational vaccines and placebo had two vaccination regimens, a single dose on day 0 or repeat dose on day 0 and 30. Among each cohort, 2 sentinels (*n* = 1 active; *n* = 1 placebo) were be assigned for a safety observation. In the absence of clinically significant safety signals in sentinel participants over a minimum period of 24 h following vaccination, the remaining participants in the cohort could be vaccinated in a sequential manner. Enrolment into high-dose groups occurred only after a safety monitoring committee reviewed the data following vaccination of the participants in the previous low-dose group. Participants received vaccinations via an intramuscular injection with RSV-G/CsA reconstituted solution or placebo according to a single (at day 0) or repeat (at days 0 and 30) vaccination schedule, with follow-up occurring for 60 days (all recipients) and 90 days (repeat dose recipients only) after the last vaccination.

### 2.2. The Vaccine

Advaccine Biopharmaceuticals Suzhou Co. Ltd., (Suzhou, China) in China manufactured the lyophilized powder of RSV-G and CsA diluent. The formulation buffer without active components was used as placebo. RSV-G lyophilized powder and vaccine diluent sterile solution were mixed together as the active BARS13 vaccine for injection. The information on study vaccine lots is listed in Appendix A.

### 2.3. Ethical Compliance

This study was conducted in accordance with the principles of ICH-GCP, the Declaration of Helsinki, and applicable local regulations for conducting clinical trials on human medicinal products. This protocol was approved by the Alfred Hospital Ethics Committee. Human sera and PBMC were prepared in 360Biolabs in Melbourne, Australia. Immunological tests were performed in Agilex Biolabs in Brisbane, Australia (anti-RSV G protein IgG antibody and neutralizing antibody) and Advaccine Biolabs in Suzhou, China (multiple cytokines assay and CD4+ T cell proliferation test).

### 2.4. Adverse Events

The severity and relationship of adverse events (AEs) to the vaccine regimens were assessed by the investigators based on the US Food and Drug Administration (FDA) standards (FDA 2007, Guidance for Industry: Toxicity Grading Scale for Health Adult and Adolescent Volunteers Enrolled in Preventive Vaccine Clinical Trials). A placebo group was included in each cohort to serve as a comparative set that would facilitate the assessment of AEs potentially caused by the vaccine. Investigators were blinded to treatment assignment during the study to maintain unbiased assessment of AEs. The study participants were issued a daily diary card to capture treatment-emergent adverse events (arthralgia, diarrhea, fatigue, fever, headache, myalgia, injection site pain, swelling, and redness) during the 30-day follow-up period after each vaccination. Chemistry, hematology, and urinalysis were assessed using clinical samples (blood and urine) collected pre-vaccination on days 0 and 30, and days 7 and 30 after each vaccination. Vital signs were measured at 30 and 60 min before vaccination, and 7 days following each vaccination. Abnormal indicators of laboratory tests and vital signs were collected as AEs if accessed to be clinically significant by the principal investigator.

The safety of BARS13 treatment regimens was based on the induction of adverse events (AEs) that includes both clinical and laboratory evaluations, using criteria that were pre-specified in the study protocol. We recorded the solicited treatment-emergent adverse events (TEAEs) during the first 30 days’ safety observation period after each vaccination, including the 30 min safety observation period after each vaccination and study days from 0 to 30 and from 30 to 60. Serious AEs (SAEs) reported from day 0 to the last study visit were also included in the primary safety endpoint.

### 2.5. Safety Data Analysis

Since this is a pilot study, the sample size was determined based on practical and logistical considerations. A sample size of 60 participants was considered appropriate to achieve the defined objectives for the study. The safety population included all participants who received any treatment (BARS13 or placebo). Basic descriptive analysis was used for safety and tolerability data. The per-protocol (PP) population consisted of all participants in the immunogenicity population who received all treatments without any major protocol deviations.

Within the PP set, the incidence of TEAEs, along with the ≥8% incidence rate of localized TEAEs and systematic TEAEs post each vaccination, were presented.

### 2.6. Determination of Anti-RSV G Protein IgG Antibodies with ELISA Assay

Serum samples collected from all participants enrolled in the study on day 0 before vaccination and days 30 and 60 post vaccination were used for anti-RSV G protein IgG antibodies evaluation using a validated sandwich ELISA assay. Plates were coated with the rRSV protein G, followed by blocking. A standard RSV IgG serum (NIBSC, London, UK, Cat No.: 16/284) was serially diluted to set the standard curve ranging from 0.156 to 10.0 IU/mL. Human serum samples were diluted (at MRD of 1 in 1000) and added to the plate for 1 h incubation. After washes, goat anti-human IgG (H+L) peroxidase-labeled anti-protein G IgG antibodies (Invitrogen, Carlsbad, CA, USA, Cat No.: 31410) were subsequently applied to the plate for 1 h incubation, followed by washes. A colorimetric signal was developed by the addition of TMB (Sigma-Adlrich, St. Louis, MO, USA, Cat No.: T0440) and a stop solution. The signal was read on an ELISA plate reader (SpectraMax VersaMax, Molecular Devices, Sunnyvale, CA, USA). The signal produced was proportional to the amount of analyte present and interpolated from the calibration curve presented on each plate. The concentrations of anti-RSV G protein IgG antibodies in the samples were determined automatically by software SoftMax Pro (Molecular Devices, Sunnyvale, CA, USA, version 7.1) by reading the calibration curves (4-PL curve fitting with 1/Y weighting factor). The data were then exported to Microsoft Excel (version 2021) and GraphPad Prism (GraphPad, San Diego, CA, USA, version 9.3) for further analysis.

All participants enrolled in the study were seropositive at baseline, showing a detectable level of anti-RSV-G IgG in their blood samples prior to the BARS13 administrated. Consequently, calculations of seropositivity and seroconversion rates were redundant. Due to this reason, the highest plasma dilution at which anti-RSV G protein antibodies were still detectable in the ELISA assay showed similar results at all assessed time points, including baseline (day 0 before vaccination) and days 30 and 60 per immunogenicity population in this study. Therefore, it was decided to evaluate the humoral response at a serum dilution equal to 1:2000.

### 2.7. Ligand Binding Assay

The neutralization effect of the RSV protein G vaccine on the RSV infection was evaluated via an anti-CCD IgG ELISA assay. The RSV envelope G glycoprotein contains a ~40 amino acid central conserved domain (CCD; amino acids 162~196) that lacks glycosylation and plays a critical role in virus infection and pathogenesis. RSV G CCD contains a CX3C motif that facilitates binding to the CX3CR1 receptor, leading to an RSV infection in human airway epithelial cells. A previous study has shown that RSV G CCD is an exposed region that is accessible to antibody binding, and the antibody against this region could exhibit strain independence and neutralize the RSV infection of human airway epithelial cells. Before evaluating the direct ELISA assay, we tried to use CX3CR1-positive human airway epithelial cells (pHAECs) to develop an assay for RSV-neutralizing antibodies. However, the results of this assay, after validation and comparison with the ELISA assay, were considered to have a lower match with the trend of neutralizing antibodies of BARS13 (unpublished data, manuscript in preparation). Herein, a ligand binding assay (LBA) was developed to detect the potential neutralization antibody level within the serum of participants. The assay format was similar to the Anti-RSV G Protein IgG Antibodies ELISA Assay. Plates were coated with RSV G CCD peptide amino acid 162~196. A standard curve was normalized using NIBSC standard RSV IgG serum (Cat No.: 16/284) and ranged from 0.60 to 75.00 IU/mL. Human serum samples were diluted (at MRD of 1 in 200) and added to the plate. HRP Anti-Human IgG (Clone: G18-145) was applied as a detecting antibody. The plates were read at 450 nm and 620 nm on a VersaMax plate reader.

### 2.8. CD4+ T Cell Proliferation Tested with the Flow Cytometry Method

Anticoagulant peripheral blood samples collected from all participants in LDR and HDR on day 0 before vaccination, day 7 post first vaccination, and days 7 and 30 post the second vaccination and lymphocytes were separated by Ficoll-plaque (Cityva, Logan, UT, USA) and cryo-frozen in liquid nitrogen for long-term storage. When lymphocytes were used for CD4+ T cell proliferation evaluation with the flow cytometry method, the cells revived from liquid nitrogen tanks were assessed using the live/dead ratio and counted. Cells at 1 × 10^6^ for each well in a 96-well plate were cultured in a cell incubator at 37 °C with 5% CO_2_ for 120 h and then stimulated by 100 μL CD3/CD28 beads per well (Gibco, Grand Island, NE, USA, Cat No.: 11131D) as a positive control, 2 μg of RSV G peptide pools per well as antigen-specific stimulation, and 100 μL of M solution (RPMI1640 spiked with 40 ng of human-IL-2 (Peprotech, Cranbury, NJ, USA, Cat No.: AF-200-02) and 40 ng of human-CD28 (MiltenyiBiotech, Bergesch Gladbach, Germany, Cat No.: 130-093-375)) as a negation control, respectively, for 120 h in vitro incubation in a cell incubator at 37 °C with 5% CO_2_. These cells were stained with anti-human CD4-AF700 (Invitrogen, Cat No.: 5600488Z)/Fixable Viability Dye-eflour 780 (Invitrogen, Cat No.: 650865514) for 30 min, fixed and permeabilized with Fixation/Perm Diluent (Invitrogen, Cat No.: 00-5223-56/00-8333-56), intra-cellular staining with anti-human Ki67-BV421 (BD Bioscience, Franklin Lakes, NJ, USA, Cat No.: 562899) for 1 h, and then applied for data acquisition on a flow cytometer (Attune NxT, Invitrogen, Carlsbad, CA, USA). The data were analyzed using FlowJo (BD Bioscience, version 10.6). Percentage of Ki67-positive cells gated from living CD4+ lymphocytes represented the proliferation of CD4+ T cells.

### 2.9. Multiple Cytokines Assay with Beads Based on the Flow Cytometry Method

Anticoagulant peripheral blood samples collected from all participants in LDR and HDR on day 0 before vaccination, day 7 post first vaccination, and days 7 and 30 post the second vaccination were treated similarly as in the above T cell proliferation assay to generate the counted revived lymphocytes. Cells at 1 × 10^6^ per well in a 96-well plate were cultured in a cell incubator at 37 °C with 5% CO_2_ overnight (12 to 16 h) and then stimulated by 100 μL CD3/CD28 beads per well (Gibco, Cat No.: 11131D) as a positive control, 2 μg of RSV G peptide pools per well as antigen-specific stimulation, respectively, for 24 h in vitro incubation in a cell incubator at 37 °C with 5% CO_2_. Cell culture supernatants were collected and reacted with a commercial cytokine detection kit (Human Th Cytokine Panel with V-bottom Plate (Biolegend, San Diego, CA, USA, Cat No.:741028)) for secreting cytokines (including IFN-γ, TNF-α, and IL-4) analysis. The data were acquired on a flow cytometer (Attune NxT, Invitrogen) and analyzed using LEGENDplex Software (Biolegend, version 8).

### 2.10. Statistical Analysis

No statistical hypothesis was formulated for this study, and only descriptive statistics was performed for the safety data as a primary outcome. Immunogenicity figures were plotted using GraphPad Prism 9. To delineate the geometric mean concentration (GMC) differences of binding antibody responses between each cohorts, a Kruskal–Wallis test (H test) was performed for GMFI levels among cohorts. In addition, to estimate the difference between and inside the cohorts for the GMFI as a secondary outcome, a post hoc analysis was performed using the Kruskal–Wallis test (H test). All statistical tests were two-sided, and differences with a *p* < 0.05 were considered significant.

## 3. Results

### 3.1. Study Design

A total of 92 participants were screened for enrollment in this trial. Among them, 32 participants were excluded, and 60 eligible participants were enrolled and randomized. All participants received vaccination by BARS13 or placebo and hence were included in the safety population. Fifty-six (93.3%) were included in the immunogenicity population, and 53 (88.3%) were included in the per-protocol population (Figure 1).

The majority of participants were white females with 46 (76.7%), and 14 (23.3%) were males. Demographics and baseline characteristics were comparable between participants vaccinated with BARS13 or placebo across all cohorts (Table 1).

### 3.2. Vaccine Safety

No SAE was experienced by any of the study participant at any time during the study. No TEAEs were classified as severe or life-threatening. No TEAE leading to study withdrawal during the 30-day follow up period after vaccination except one TEAE of moderate asthma exacerbation reported by a placebo participant in LDR. The majority of the TEAEs recorded were classified as mild. The frequency of TEAEs and drug-related TEAEs did not increase with vaccine dose level and frequency. The figures of the overview AEs post each vaccination were attached in Appendix A.

Local pain/tenderness was the most frequent solicited local adverse reaction in participants treated with the active vaccine. Fatigue was the most frequently reported solicited systemic adverse reaction, and it was the most frequently reported as severe. The incidence rate of adverse reactions did not increase with vaccine dose level and frequency. Furthermore, the incidence rates of most local and systemic adverse reactions showed a detectable decrease after the second vaccination at day 30 compared with those after the first vaccination at day 0, independently of the vaccine dose.

The majority of solicited adverse reactions were classified as mild, and none were classified as life-threatening. After the first vaccination on day 0, the most frequent localized adverse reaction in 24 low-dose recipients (low-dose single and low-dose repeat groups) and 24 high-dose recipients (high-dose single and high-dose repeat groups) was localized pain/tenderness, with the incidence rates of 45.8% and 66.7%, respectively (Figure 2a). Five (20.8%) events of localized pain/tenderness were reported as moderate in high-dose recipients. In 12 placebo recipients, localized pain/tenderness (8.3%) was also reported after the first vaccination. Other localized adverse reactions were reported no more than 8.3% in all cohorts. After the second vaccination on day 30, three localized adverse reactions were reported among 12 LDR recipients, 12 HDR recipients, and six placebo recipients. Two (16.7%) moderate localized pain/tenderness events were reported in LDR and HDR recipients, respectively. Two (16.7%) mild localized pain/tenderness events were reported in LDR and HDR recipients, respectively. One (16.7%) mild localized pain/tenderness event and one (16.7%) mild ecchymosis/discoloration event were reported in placebo recipients. Meanwhile, ecchymosis/discoloration and swelling/induration in LDR and HDR recipients were reported in no more than 8.3% after the second vaccination.

After the first vaccination on day 0, the most frequent systematic adverse reactions were fatigue (41.7%), headache (20.8%), myalgia (16.7%), and malaise (16.7%) in low-dose recipients (Figure 2b). Among high-dose recipients, the most frequent systematic adverse reactions were fatigue (33.3%), myalgia (29.2%), headache (16.7%), lightheadedness (16.7%), and malaise (12.5%). The incidences of fatigue, headache, and malaise were relatively high among placebo recipients, with five (41.7%), seven (58.3%), and three (25.0%) events reported, respectively. Compared with all the mild and moderate fatigue and headache events reported in both low- and high-dose recipients, one (8.3%) event of fatigue and one (8.3%) headache were each reported as severe among placebo recipients. No severe fatigue was reported among both low- and high-dose recipients. After the second vaccination on day 30, one (8.3%) fatigue event was reported among HDR recipients. Other systematic adverse reactions were reported as mild in all cohorts.

### 3.3. Specific G Protein-Binding Antibody Response

In the immunogenicity and per-protocol populations, the value of concentrations and the GMFI of G protein-binding antibodies in terms of the BARS13 dosed cohorts on days 30 and 60 (repeat-dose regimen only) were numerically higher with those on day 0 in anti-RSV-G IgG ELISA absorbance values using the collected serum at 1:2000 dilution.

The antibody concentrations at days 0, 30, and 60 in low-dose recipients (Figure 3a) and high-dose recipients (Figure 3b) each compared with those in placebo recipients were presented. The GMCs at day 30 were 1049.08 IU/mL (95% CI: 519.94–2116.72) and 1126.61 IU/mL (95% CI: 624.65–2031.94) for LDS and HDS participants, respectively. The GMCs of LDR and HDR participants on day 30 were 763.18 IU/mL (95% CI: 380.59–1530.36) and 885.74 IU/mL (95% CI: 406.25–1931.17), respectively. On day 60, GMCs among LDR and HDR participants were both higher than baseline, 1187.10 IU/mL (95% CI: 610.01–2310.13) and 1482.12 IU/mL (95% CI: 706.56–3108.99), respectively. From the distribution of antibody concentrations, after receiving one or two doses of BARS13, an apparent upward trend in the antibody concentration of all vaccine recipients was observed. Especially for participants who had a two-dose regimen (LDR and HDR), the increases of their IgG antibody concentrations were much higher at day 60 than at day 30.

The antibody data analysis demonstrated that the levels of anti-G antibodies elicited in all BARS13 recipients were superior to those of the placebo recipients at all sampling time points. In terms of the dosage–effect relationship between times of vaccinations and GMCs, LDR and HDR participants who received the two-dose regimen showed a detectable increase in the concentration of binding antibodies at day 60 when compared with that of LDS and HDS participants who received a single-dose regimen, suggesting that the two-dose regimen was more advantageous in terms of generating more binding antibodies against RSV. In terms of dose selection, the high-dose cohorts (HDS and HDR) showed higher anti-G antibody concentrations on days 30 and 60 (LDR and HDR only) than the low-dose groups did (LDS and LDR), respectively. Based on the analysis of the GMCs of BARS13 binding antibodies, the increase of binding antibodies was positively correlated with the increased dose and dosage of BARS13.

The GMFIs at days 0, 30, and 60 in low-dose recipients (Figure 3c) and high-dose recipients (Figure 3d), each compared those in with placebo recipients, are presented. Descriptively, the GMFIs for LDS and HDS participants with the BARS13 single dose at day 30 were 1.72 (95% CI: 1.23–2.41) and 1.75 (95% CI: 1.34–2.29), respectively, both higher than that of the placebo recipients with a GMFI of 1.01 (95% CI: 0.92–1.11) at day 30. The GMFIs for LDR and HDR participants with the BARS13 two-dose regimen at day 30 were 2.04 (95% CI: 1.44–2.88) and 1.89 (95% CI: 1.36–2.63), while at day 60 they were 3.17 (95% CI: 1.88–5.36) and 3.16 (95% CI: 1.99–5.03), respectively. Comparatively, the GMFI for participants who received placebo at day 60 was much lower with the increase-fold down to 0.96 (95% CI: 0.91–1.01).

### 3.4. Anti-CCD Antibody Response

Per the immunogenicity population, the anti-CCD IgG antibody level, which is likely to represent the neutralization antibody level, at days 0, 30, and 60 in low-dose recipients (Figure 4a) and high-dose recipients (Figure 4b), each compared with that in placebo recipients, are presented. The increase of neutralization antibody concentrations in HDS and HDR was observed 30 days after their last vaccination. The GMC value of neutralization antibodies in LDR and HDR at day 30 was 1161.1 IU/mL (95% CI: 599.6–2248.4 IU/mL) and 1507.9 IU/mL (95% CI: 1.8997–1.3637 IU/mL), respectively, showing a detectable high increase compared with the baseline GMC value. At day 60, the GMC value of the neutralization antibodies maintained an increasing trend and was up to 1683.6 IU/mL (95% CI: 904.1–3135.4 IU/mL) and 2499.6 IU/mL (95% CI: 1195–5228.3 IU/mL), respectively.

The GMC values in LDS, HDS, and participants dosed with placebo at day 0 was 791.2 IU/mL (95% CI: 476.6–1313.6 IU/mL), 1040.9 IU/mL (95% CI: 618.9–1750.8 IU/mL), and 1004.2 IU/mL (95% CI: 552.7–1824.5 IU/mL), respectively. The GMC values with the 95% CI of neutralization antibodies in LDS, HDS, and participants dosed with placebo at day 30 was 1163.7 IU/mL (95% CI: 725.1–1867.4 IU/mL), 1595.3 IU/mL (95% CI: 964.7–2638.2 IU/mL), and 1307 IU/mL (95% CI: 259.3–2764.3 IU/mL), respectively. The GMC value of neutralization antibodies in participants dosed with placebo on days 0, 30, and 60 was 1004.2 IU/mL (95% CI: 552.7–1824.5 IU/mL), 1307 IU/mL (95% CI: 259.3–2764.3 IU/mL), and 846.6 IU/mL (95% CI: 259.3–2764.3 IU/mL), respectively.

### 3.5. Cellular Response

Having been demonstrated in animal studies, BARS13 immunizations would induce Tregs that could suppress T cell proliferations when animals were exposed to RSV infection. To test if this is also true in a human setting, we set up a flow cytometry method to explore proliferative profiles and functions of T cells being restimulated in vitro by the RSV G peptide from LDR and HDR groups vaccinated by BARS13 in this trial. This test has been done in a post hoc setting, hence subjects were unblinded and only subjects receiving BARS13 were included.

In LDR, compared with the high response readout stimulated by CD3/CD28 as positive stimulants, the levels of G peptide-stimulated IFN-γ and Ki67 were relatively stable with a minimal increase after the second vaccination at days 37 and 60, while the level of G peptide-stimulated TNF-α showed no obvious change from days 0 to 30 and actually decreased at days 37 and 60. The level of IL-4 stimulated by G peptide was lower than that stimulated by CD3/CD28. The median and quartile values of IFN-γ stimulated by G peptide at days 0, 7, 30, 37, and 60 were 6.13 (2.33, 10.97), 1.93 (1.52, 3.35), 6.32 (6.32, 6.32), 1.78 (1.78, 1.78), and 1.25 (1.12, 3.99) pg/mL, respectively (Figure 5a).

In the HDR group, all other Th1-related cellular cytokines (IFN-γ, TNF-α, and Ki67) showed no obvious change from pre-vaccination to post-vaccination timepoints for the G peptide-stimulated samples. Comparatively, CD3/CD28-stimulated samples showed higher responses to the aforementioned cytokines. For Th2-biased cytokine IL-4, a consistent pattern with that of LDR also showed that positive stimulant samples also generated a numerically higher readout across all timepoints (Figure 5b).

## 4. Discussion

We have performed a first-in-human phase I trial on BARS13, a novel designed RSV recombinant G protein vaccine with an immunomodulator, CsA. This RSV vaccine candidate was designed with the aim to suppress over-reactive T cells related to VED risk that has been observed in previous RSV vaccine clinical programs [25,26]. Before this phase I trial, BARS13 vaccinations have shown that RSV binding and neutralizing antibodies can be significantly increased, and no VED symptom has been observed following detailed histopathology examination in a murine model RSV challenge study. This suppressed cellular response was associated with Tregs induction since Treg-knocked animals lost the ability to prevent VED in the same challenge study [25]. In a rabbit study, BARS13 vaccinations could induce long durable and recalled anti-RSV G antibody responses, but without T cell proliferations after being stimulated by the G peptide in vitro [26].

In this phase I trial, the majority of solicited local adverse reactions were classified as mild, and none as severe nor life-threatening. No clinically significant vaccine-related safety or tolerability signals were reported during this study. The administration of BARS13 was generally tolerable, with no apparent differences between BARS13- and placebo-vaccinated participants. The first-in-human study of BARS13 showed a tolerable and promising safety profile for this RSV vaccine candidate.

In the immunogenicity investigations, the anti-RSV-G IgG antibody concentrations measured by ELISA were expressed as the concentration change from baseline and GMFI from baseline with 95% CIs for each of the individual treatment groups on days 30 and 60. The antibody levels as well as the fold increases from baseline to post vaccination indicated a dose-dependent increase pattern from the low-dose to high-dose cohorts. The boost dose also contributed to the antibody response as shown in repeat-dose cohorts with the binding antibody level having increased further after the second dose.

Previous studies have shown that vaccine-enhanced disease (VED) was due to over-reactive CD4+ T cells, but it is not clear how different CD4+ T cell subsets lead to increased risk of VED [27]. In the meantime, human airway epithelial cells and alveolar macrophages can produce proinflammatory cytokines, such as tumor necrosis factor-α (TNF-α), which can help stimulate immune responses to inhibit RSV infection. Similarly to children immunized with FI-RSV, BALB/C mice experimentally exhibited VED associated with Th2-biased immune responses [28]. In order to explore whether BARS13 vaccination will lead to a proinflammatory cellular response, Th1-type cytokines (IFN-γ and TNF-α) and Th2-type cytokines (IL-4) were assessed by flow cytometry to observe the T cell immune response induced by BARS13 vaccination. From the PBMC testing results, it could be shown that the G peptide-stimulated PBMCs generally did not show an obvious response to stimulation, but comparatively, anti-CD3/CD28 positive stimulant samples exhibited a significant response to the in vitro stimulation, demonstrating the responsiveness of PBMCs to an external stimulant and potentially leading to the conclusion that BARS13 vaccination does not induce over-reactive T cells and has a lower chance to develop VED once RSV re-exposure occurs in those vaccinated subjects.

There are some limitations to this phase I study. First, the limited follow-up period may not allow sufficient observation of potentially delayed reactions. Second, utilizing CX3CR1-positive primary human airway epithelial cells (pHAECs), we sought to develop an RSV neutralizing antibody assay. However, the assay’s results were inconclusive due to the inter-batch variability of primary culture cells. To address this challenge, we developed a new method, LBA, to measure G protein-based neutralization based on our extensive comparative studies (unpublished data, manuscript in preparation), and the test data, which were likely to be considered as neutralization data from the LBA method, were finally presented after validation (Figure 4). Despite this, additional optimization of the RSV G protein-based neutralization procedure is required for use in the future. Third, due to extended long cryopreservation time of the collected PBMCs (more than 1 year), successful retrieval of cell samples was variable and rendered the result interpretations of flow analysis to be only tentative.

## 5. Conclusions

In summary, the first-in-human trial has demonstrated that BARS13 not only induced a meaningful level of anti-RSV-G Abs in a dose-dependent fashion, it also, importantly, demonstrated a well-tolerable and excellent safety profile from the recombinant RSV G protein with a low concentration of CsA in the formulation. This novel adjuvant, CsA, provides the potential of circumventing enhanced respiratory disease in the history of RSV vaccine development, which warrants further exploration in future clinical trials.

## Figures and Tables

**Figure 1 vaccines-11-00999-f001:**
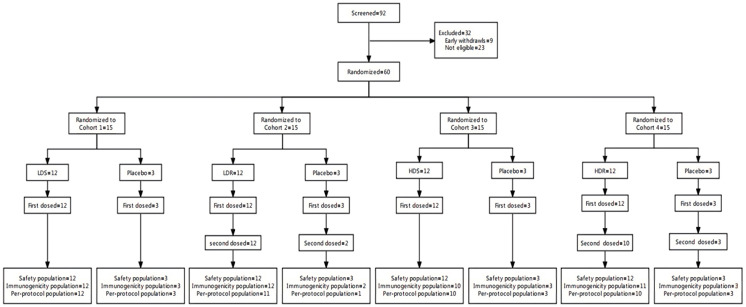
Study profile. In each cohort, 2 sentinels (*n* = 1 active; *n* = 1 placebo) were assigned for a safety observation at the study site for a minimum period of 60 min post vaccination. Upon completion of the on-site safety assessments and release from the site, the sentinel participants were monitored by follow-up telephone calls (at least one call) over a minimum period of 24 h following vaccination. In the absence of clinically significant safety signals in sentinel participants over this period, the remaining participants in the cohort could be vaccinated in a sequential manner, with a minimum interval between participants of 30 min to allow monitoring of any acute events. During the vaccination period, the 7-day safety data of Cohort 1 (includes low-dose single receipts and placebo receipts) will be reviewed by the safety review committee (SRC) if no safety concern has raised, and the initiation of enrolment for both Cohort 2 (includes low-dose repeat receipts and placebo receipts) and Cohort 3 (includes high-dose single receipts and placebo receipts) (Step II; after review of Cohort 1 data) and Cohort 4 (includes high-dose repeat receipts and placebo receipts) (Step III; after review of Cohort 3 data) could be triggered. One placebo receipt from Cohort 2 and 2 BARS13 recipients from Cohort 4 failed to complete the second vaccination on day 30, leading to exclusion in the per-population set. LDS: low-dose single. LDR: low-dose repeat. HDS: high-dose single. HDR: high-dose repeat.

**Figure 2 vaccines-11-00999-f002:**
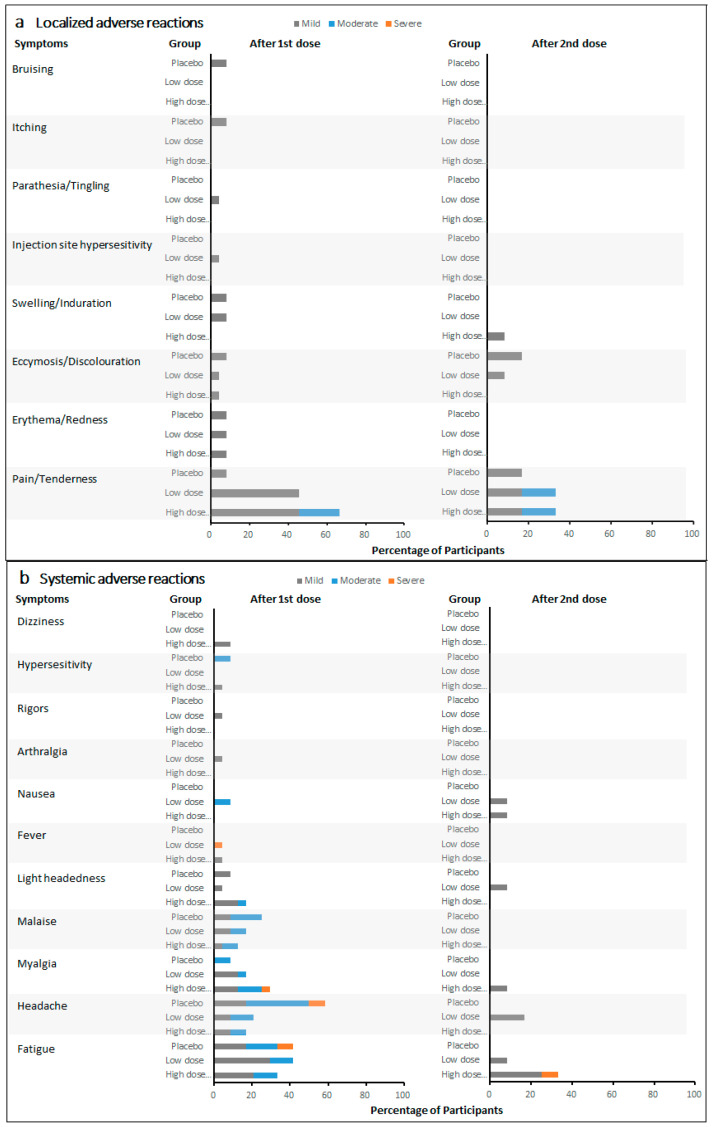
Incidence of localized and systematic adverse reactions in all cohorts after each vaccination. (**a**) Incidence of localized adverse reactions after each vaccination. (**b**) Incidence of systemic adverse reactions after each vaccination.

**Figure 3 vaccines-11-00999-f003:**
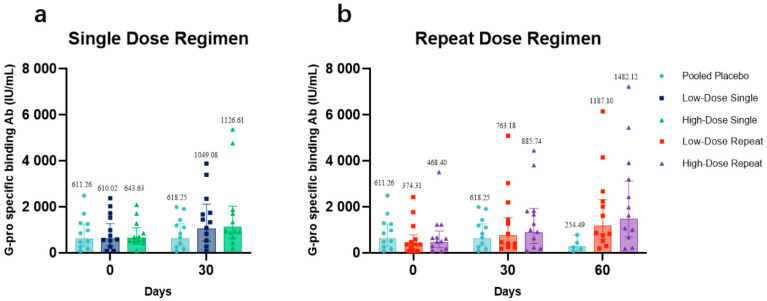
G protein-binding antibodies in all cohorts. (**a**) The geometric mean concentrations (GMCs) of binding antibodies in low-dose recipients compared with those in placebo recipients. (**b**) The geometric mean concentrations (GMCs) of binding antibodies in high-dose recipients compared with those in placebo recipients. With the ELISA test method, the concentrations of anti-RSV G protein antibodies in the plasma at days 0, 30, and 60 for high/low-dose and placebo cohorts were quantified using a standard curve prepared with the standard product antiserum to RSV. The original concentration was 2000 IU/mL. GMC and 95% confidence intervals were obtained from GraphPad Prism 9. (**c**) The geometric mean fold increase (GMFI) of binding antibodies in high-dose recipients compared with that in placebo recipients. (**d**) The geometric mean fold increase (GMFI) of binding antibodies in high-dose recipients compared with that in placebo recipients. The GMFI of BARS13 IgG antibody level in all cohorts at days 0, 30, and 60 (LDR and HDR only) and 95% confidence intervals were obtained from GraphPad Prism 9. *p*-values were tested using the Kruskal–Wallis testing method (** *p* ≤ 0.01, * *p* ≤ 0.05, ns represents *p* > 0.05).

**Figure 4 vaccines-11-00999-f004:**
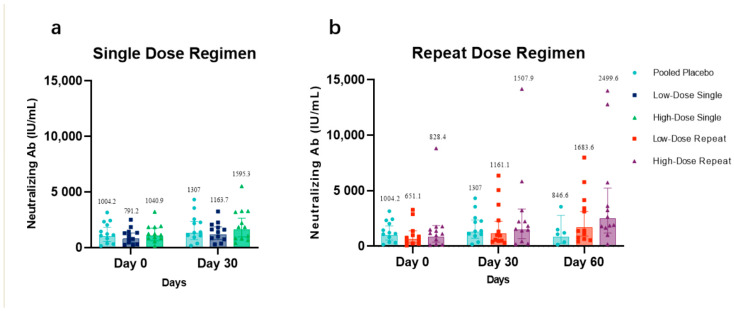
Potential neutralizing antibody levels tested with LBA in all cohorts. (**a**) The geometric mean concentrations (GMCs) of anti-CCD antibodies (likely to be neutralizing antibody) in low-dose recipients compared with those in placebo recipients. The GMCs of anti-CCD antibodies of single-dose regimens and placebo recipients at days 0, 30, and 60. Missing data were imputed using the last observation carried forward (LOCF) method. (**b**) The geometric mean concentrations (GMCs) of anti-CCD antibodies in high-dose recipients compared with those in placebo recipients. The GMCs of anti-CCD antibodies of repeat-dose regimens and placebo recipients at days 0, 30, and 60.

**Figure 5 vaccines-11-00999-f005:**
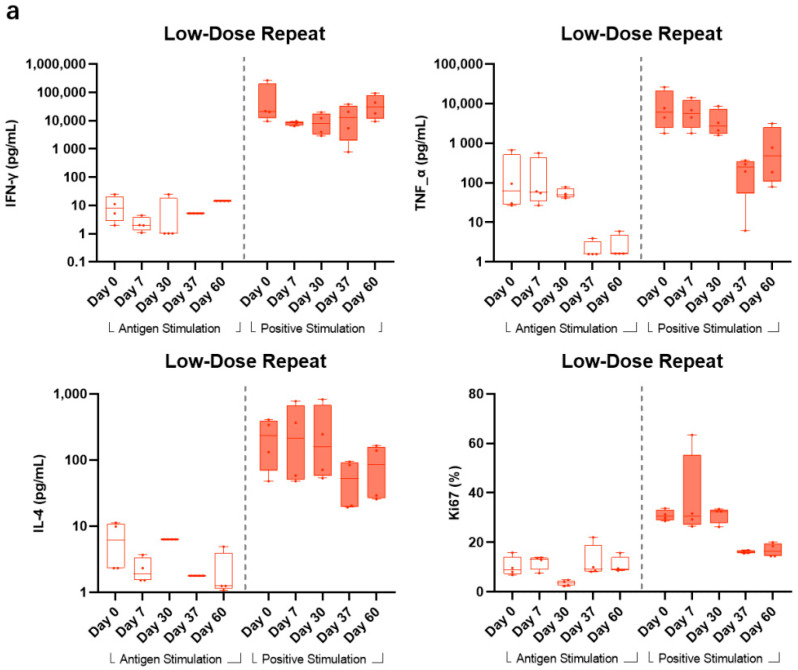
Specific T cell response. (**a**) T cell response in low-dose-repeat recipients. The median and quartile values of Th1 cytokines (IFN-γ and TNF-α), Th2 cytokine (IL-4), and Ki67 in LDR were detected at days 0, 7, 30, 37, and 60 with the stimulation of G peptide as the specific antigen and CD3/CD28 as a positive control on cryopreserved PBMCs. Due to some samples on day 0 not recovering successfully from the long (>1 year) cryopreservation, several subjects’ samples’ testing results lacked baseline control, and a decision was made that only ≥4 BARS13 recipients with their PBMC samples available at days 0, 7, 30, 37, and 60 in each cohort were included in the cellular response analysis. Therefore, 4 (33.4%) BARS13-reciptients in LDR were included in the T cell immunogenicity analysis. For the percentage readout of Ki67, a total of 11 (91.7%) BARS13 LDR recipients were available for the analysis. (**b**) T cell response in high-dose-repeat recipients. Th1 cytokines (IFN-γ and TNF-α), Th2 cytokine (IL-4), and Ki67 in HDR were detected at days 0, 7, 30, 37, and 60 with the stimulation of G peptide as the specific antigen and CD3/CD28 as a positive control on cryopreserved PBMCs. Four (33.4%) BARS13 HDR recipients were included in the T cell immunogenicity analysis. All 12 BARS13 HDR recipients were included in the Ki67 proliferation analysis.

**Table 1 vaccines-11-00999-t001:** Demographics and baseline characteristics (all participants’ population).

Characteristics	LDS (N = 12)	LDR (N = 12)	HDS (N = 12)	HDR (N = 12)	Pooled Placebo (N = 12)	Overall (N = 60)
Age (years)	25.40 (3.99)	27.30 (6.72)	28.80 (7.16)	27.60 (6.76)	28.00 (2.22)	27.40 (5.87)
BMI at screening (kg/m^2^)	25.44 (4.86)	24.11 (5.39)	24.21 (4.25)	23.97 (3.21)	24.65 (3.77)	24.48 (4.73)
**Gender**						
Female	11 (91.7%)	10 (83.3%)	8 (66.7%)	9 (75.0%)	8 (66.7%)	46 (76.7%)
Male	1 (8.3%)	2 (16.7%)	4 (33.3%)	3 (25.0%)	4 (33.3%)	14 (23.3%)
**Ethnicity**						
Hispanic or Latino	1 (8.3%)	2 (16.7%)	1 (8.3%)	1 (8.3%)	0 (0.0%)	5 (8.3%)
Not Hispanic or Latino	11 (91.7%)	10 (83.3%)	11 (91.7%)	11 (91.7%)	12 (100.0%)	55 (91.7%)

Data are mean (SD) or n (%).

## Data Availability

The data generated or analyzed during this study are included in this published article and its Appendix A.

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
