# Peer review of "A First-in-Human Trial to Evaluate the Safety and Immunogenicity of a G Protein-Based Recombinant Respiratory Syncytial Virus Vaccine in Healthy Adults 18–45 Years of Age"

_vaccines, 2023, doi:10.3390/vaccines11050999_

Round 1
Reviewer 1 Report
Vaccines RSV paper
The authors describe results from a first in humans clinical trial of a novel vaccine for respiratory syncytial virus (RSV), BARS13. The vaccine, which showed promising results in small animal models, contains the RSV G protein along with an immunomodulatory agent, cyclosporine A (CsA). Vaccination of 60 healthy adults gave no serious safety signals and the vaccine was well tolerated. There was a modest, but not significant, increase in antibody responses to the RSV glycoprotein G.
Specific comments:
1. While that paper is generally well written there are a few sentences that need editing for clarity. The paper would benefit by a thorough editing for language.
2. The biggest issue with this report is the use of a non-standard, surrogate assay for virus neutralization. Both assays described in the Methods section are measuring binding antibodies to RSV G. Given the smaller number of study subjects and the modest results from the non-standard assay, the authors should consider preforming a standard virus neutralization assay.
3. Serologic boosting is usually defined as a four-fold increase in antibody titer. Therefore, the overall three-fold boost reported here is not significant.
4. The RSV G antigen is expressed in E. coli. Have the authors studied the antigenic characteristics of this expressed antigen? Bacteria usually don’t do a good job of expressing mammalian glycoproteins in a native configuration. This could lead to the production of binding antibodies, that may not recognize conformational epitopes on the native protein. Recognition of native epitopes may be required for efficient neutralization.
Author Response
Dear Reviewer,
Thanks for the opportunity to allow us to modify our manuscript entitled: “A First-in-Human Trial to Evaluate the Safety and Immunogenicity of a G Protein Based Recombinant Respiratory Syncytial Virus Vaccine in Healthy Adults 18-45 Years of Age”.
In the revision, we modified, corrected and revised our manuscript according to your professional comments and suggestion. We also fully addressed the comments made by you point-to-point. I believed that this revised manuscript has been strengthen, and clarified.
Thank you again for the suggestions!
Best Regards

Reviewer 2 Report
See attached file.

Author Response
Dear Reviewer,
Thanks for the opportunity to allow us to modify our manuscript entitled: “A First-in-Human Trial to Evaluate the Safety and Immunogenicity of a G Protein Based Recombinant Respiratory Syncytial Virus Vaccine in Healthy Adults 18-45 Years of Age”.
In the revision, we modified, corrected and revised our manuscript according to your professional comments and suggestion. We also fully addressed the comments made by you point-to-point. I believed that this revised manuscript has been strengthen, and clarified.
Thank you again for your suggestions!
Best Regards

Round 2
Reviewer 2 Report
The authors have addressed all my queries successfully and in detail, and they have amended the manuscript accordingly.
My only suggsestion is that regarding the comment 6, they should add a very brief discussion of the Ligand binding assay (LBA) and cite it as "unpublished data, manuscript under preparation".
Author Response
Dear Reviewer,
We would like to thank you again for the opportunity to allow us to further modify our manuscript. In this revision, we revised our manuscript according to your thoughtful suggestion. I believed that this revised manuscript, especially for the discussion section, has been more scientific and rigorous.
We are very pleased to have all your suggestions for this manuscript, which are very valuable to us.
Thanks & Best Regards
The authors,
Response to Reviewer 2 Comments
Reviewer 2. Comment 1. Round2. My only suggsestion is that regarding the comment 6, they should add a very brief discussion of the Ligand binding assay (LBA) and cite it as "unpublished data, manuscript under preparation".
Response 1: Thank you for your suggestion. The discussion of LBA as a surrogate method has been added in the discussion section with the details descriped below.(L517-L525)
We have added the contents of ‘Second, utilizing CX3CR1-positive primary human airway epithelial cells (pHAECs), we sought to develop an RSV neutralizing antibody assay. However, the assay's results were inconclusive due to the inter-batch variability of primary culture cells. To address this challenge, we developed a new method, Ligand Binding Assay (LBA), as a surrogate as-say for measuring G protein-based neutralization based on our extensive comparative studies (unpublished data, and manuscript in preparation), and the neutralization data from the LBA method was finally presented after validation (Figure 4). Despite this, addi-tional optimization of the RSV G protein-based neutralization procedure is required for use in the future.’ in the disscussion section.